# Pharmaceutical marketing and patient trust: How do doctor-targeted campaigns affect patients?

**Marta Makowska[1], Akihiko Ozaki[2,3], Piotr Ozieranski[4]***

**1** Department of Economic Psychology, Kozminski University, Warsaw, Poland, **2** Breast and Thyroid Center, Jyoban Hospital of Tokiwa Foundation, Iwaki, Japan, **3** Medical Governance Research Institute, Tokyo, Japan, **4** Department of Social and Policy Sciences, University of Bath, Bath, United Kingdom

* po239@bath.ac.uk

## Abstract

### Background

In the context of pervasive pharmaceutical marketing directed at doctors, it is crucial to understand whether patients notice these activities and, if so, what impact this may have on trust in doctor-patient relationships.

### Methods

The study was conducted through an online survey with 1,057 Polish participants. A quota sample, reflecting the Polish population in terms of specific socio-demographic characteristics, was chosen from an internet panel.

### Results

The average trust in physicians among Poles, on a 10-point scale, was 6.3 (SD = 2.1), while the average trust in pharmaceutical companies was lower, at 5.0 (SD = 2.3). The results indicate that 83.4% of respondents noticed signs of pharmaceutical marketing directed at doctors, with 5.5% experiencing all six types of marketing practices addressed in the study. Seeing a company logo in the doctor's office, encountering a pharmaceutical sales representative (PSR), and experiencing PSR-related longer waits were each associated with lower trust in physicians (t = −2.2, −2.3, −2.9; p = .028,.019,.004; d = −.136, −.148, −.188, respectively) and in pharmaceutical companies (t = −2.7, −3.1, −2.3; p = .008,.002,.021; d = −.166, −.202, −.151, respectively). Receiving a free drug sample was linked to slightly higher trust in physicians (t = 2.2, p = .028, d = .16) and showed no effect on trust in companies (p = .558). Most pairwise correlations among patient-encounter pharmaceutical marketing situations were weak, even when they reached statistical significance; the only strong association was between encountering a PSR in a medical facility and reporting PSR-related longer wait times (r = .69, p < .001).

**Data availability statement:** All relevant data can be found at the following link: https://figshare.com/articles/dataset/Transparency_Sunshine_Act_Poland/24598488.

**Funding:** This work was supported by a small grant from Kozminski University, Warsaw, Poland, awarded to MM. The funder provided support in the form of salaries for authors MM, but did not have any additional role in the study design, data collection and analysis, decision to publish, or preparation of the manuscript.

**Competing interests:** MM has nothing to declare. Consulting fees from: Beckton, Dickinson and Company, MNES Inc. Payment or honoraria for lectures, presentations, speakers bureaus, manuscript writing, or educational events from: Pfizer, Daiichi Sankyo, Taiho Pharmaceutical Co. Ltd., Kyowa Kirin. PO is former PhD student was supported by a grant from Sigma Pharmaceuticals, a UK pharmacy wholesaler and distributor (not a pharmaceutical company). The PhD work funded by Sigma Pharmaceuticals is unrelated to the subject of this paper. This does not alter our adherence to PLOS ONE policies on sharing data and materials.

## Conclusion

Physicians and pharmaceutical companies must acknowledge that their marketing relationships can influence patient trust and should carefully assess the consequences of their collaboration on the public's perception of medicine and public health.

## Introduction

Large-scale studies conducted in the United States have shown that various forms of pharmaceutical industry payments to physicians – such as sponsored meals, travel, or consulting arrangements – are significantly associated with increased prescribing of promoted products [1–3]. While European countries lack government-run payment and prescription databases enabling such research [4], such relationships are not limited to the US market-based healthcare system [5]. For example, a study of 107 family doctors in England established that pharmaceutical sales representatives (PSRs) were the top source of information about prescription drugs, with the potential to influence prescribing decisions [6]. Similar patterns have been identified in other Western European countries, including Germany [7,8] and France [9].

One main mechanism behind this influence that has been identified is generalised reciprocity – a form of moral obligation where physicians feel compelled to return 'gifts' from pharmaceutical companies by prescribing their promoted medications [10–13]. A systematic review has identified drug samples, promotional material, continuing medical education, scientific journals, and hospitality as the most frequent gifts [5]. Importantly, even small gifts, such as a free lunch, have been associated with changes in prescribing behaviour [3,14,15].

Despite the clear research evidence, only a minority of physicians acknowledge that contact with PSRs or receiving gifts influences their prescribing practices. Interestingly, they are more likely to believe that such factors affect the prescribing behaviours of their colleagues rather than their own [5,15–17]. Moreover, research indicates that some physicians even feel offended by the suggestion that industry payments could influence their decisions [18]. This disconnect between the awareness of influence and self-perception may be attributed to cognitive biases such as self-serving bias and rationalisation, which lead physicians to underestimate their susceptibility to commercial pressures [19]. For example, Hong et al. [20] indicate that patients often view gifts from pharmaceutical companies as more influential than physicians do themselves. This perception can lead to the erosion of trust in the patient-physician relationship or even loss of trust in the whole healthcare system [21,22].

Research findings often underscore the negative impact of financial relationships between physicians and the pharmaceutical industry on patients, for whom the consequences can be significant. Physicians accepting industry payments often prescribe newer, more expensive treatments rather than cheaper alternatives that are no less effective [2,3,23,24]. Payments have also been linked to the prescription of

non-recommended drugs of uncertain value [25] or treatments that are unnecessary or can be harmful because of additional risks and side effects [26,27].

In addition to these *direct* impacts on prescribing practice, increasing attention has been given to other less-explored negative consequences for patients. These *indirect* effects include longer wait times due to physicians being occupied with visits from PSRs or the receipt of potentially unbalanced marketing materials intended for physicians only.

Yet, it is worth noting that the marketing activities of the pharmaceutical industry also contain elements that may be beneficial from the perspective of the health system and patients [28,29]. Given the relative dearth of alternative funding sources, the pharmaceutical industry offers professional development opportunities, enabling physicians to regularly update their knowledge and participate in conferences and training [30–32] and, with appropriate safeguards, this education need not be biased [31]. PSRs frequently provide drug samples that can be given to patients to try a medication at no cost [30,33]. Physicians may perform various consultancies and research for industry, which constitute an important source of additional income for some doctors or contribute to the development and testing of new medicines [32,34].

Understanding how patients and the public view physician-industry interactions is crucial for shaping policies and creating targeted interventions [35,36]. Most studies to date have focused on describing 'attitudes,' 'beliefs,' and 'knowledge,' and some have also addressed attitudes towards the effect of interactions on trust [21,36,37]. However, patients' direct experiences with physician-directed pharmaceutical marketing remain relatively underexplored, particularly in Central and Eastern European contexts. The novelty of this article lies in its aim to explore Polish patients' experiences with pharmaceutical marketing targeted at physicians and analyse its implications for trust. This patient – centred perspective adds an important dimension to the existing literature, which has predominantly examined physicians' or policymakers' views. The article addresses four research questions: 1) Do Polish patients notice pharmaceutical marketing targeted at physicians? 2) Does this perception have a relationship with their trust in physicians? 3) Does noticing such marketing have a relationship with their trust in pharmaceutical companies? 4) What socio-demographic characteristics increase the likelihood of patients noticing this type of marketing?

## Theoretical framework

Patients' perceptions of marketing directed at doctors can be explored further by examining how selective attention influences what information is processed and what is ignored, particularly in the context of the doctor-patient interaction. The concept of selective attention, rooted in cognitive psychology, suggests that individuals can only process a fraction of the stimuli they are exposed to [38–40]. This means that despite being surrounded by multiple forms of information, including marketing messages, many individuals – both doctors and patients – filter out what they consider irrelevant, allowing them to focus on information that aligns with their immediate needs or interests.

The selective attention theories differ in explaining how information is prioritised [41–43]. However, the central concept of selection helps explain why some individuals notice marketing efforts while others do not, even when exposed to them. Notably, for many patients, their primary focus is not on the presence of the pharmaceutical industry in a doctor's environment but on their health and the information provided by the medical professional. As a result, the presence of pharmaceutical marketing may not register at all. The question can be posed of whether specific socio-demographic characteristics predispose individuals to notice pharmaceutical marketing directed at doctors.

Also, the phenomenon of loss of trust in physicians or healthcare, which may be associated with persuasive pharmaceutical marketing, can be understood through another concept connected with theories of selective attention – specifically, cognitive dissonance [44]. Cognitive dissonance posits that individuals experience psychological discomfort when confronted with two conflicting beliefs or when their behaviour contradicts their beliefs. In the context of selective attention, this discomfort can influence how individuals perceive and filter information. When exposed to information that challenges their existing beliefs or behaviours, individuals are prone to selectively attend to information that reinforces their current attitudes or actions while disregarding or dismissing information that creates dissonance. This process is often referred to as confirmation bias [45].

Thus, when a patient notices a pharmaceutical company logo or other marketing elements in the doctor's office, they may experience a conflict between this observation and their expectation that the doctor should always act in their best interest. To resolve this cognitive dissonance, the patient might adjust their attitudes or beliefs. One potential response is to reduce their trust in the doctor to reconcile their perception of the doctor's integrity with the presence of commercial marketing influences. This process could also trigger confirmation bias, where the patient selectively focuses on information that supports their altered view, such as instances where the doctor's actions appear commercially motivated, while disregarding evidence that might contradict this view. This reinforcement of scepticism about the doctor's objectivity can be particularly concerning in areas like vaccine distrust. Since trust is a foundational element of the patient-physician relationship, essential for effective healthcare, perceived conflicts of interest between the pharmaceutical industry and physician interactions can lead patients to question the motivations behind treatment recommendations, ultimately undermining care.

Research suggests that age is a significant factor in selective attention, with studies indicating a decline in selective attention capabilities as individuals age [39,46]. Gender has also been shown to play a role in selective attention mechanisms [47,48]. Moreover, the multifaceted responsibilities associated with having children necessitate a high degree of selective attention focused on their needs [49]. Additionally, research from China indicates that young adults who were born and raised in rural areas may process attended and unattended information differently from their urban counterparts [50]. Health status also influences selective attention; studies demonstrate that health anxiety heightens selective attention toward both internal and external health-related threats [51].

## The specificity of the Polish pharmaceutical market in the cultural context

This is the first study exploring examining systematically the effects of pharmaceutical marketing on patient trust in the context of Poland, the sixth-largest pharmaceutical market in Europe [52]. The Polish market is characterised by high levels of pharmaceutical marketing activity towards physicians, including detailing, sponsorships of medical conferences, printing educational materials, providing gifts and drug samples, research funding, advertisements in medical journals and employing doctors as speakers or consultants, etc. [15,53,54]. According to the Central Statistical Office [55], 2024 marked a record year for prescription drug sales in Poland, both in volume and value. The number of prescriptions issued rose by 3.7% compared to the previous year. Furthermore, the total value of prescription drugs purchased reached 29.6 billion PLN (≈ 6,96bn EUR; exchange rate from 3 November 2025: 1 EUR = 4.25 PLN) reflecting a year-on-year growth of 3.5 billion PLN (≈ 0,82bn EUR). This positive trend underscores the market's potential, providing strong support for the ongoing use of aggressive pharmaceutical marketing strategies.

The cultural context of pharmaceutical marketing in Poland is different from that of many Western countries, which have been the focus of most research to date. Poland has been described as a 'culture of distrust' [56], a sentiment rooted in historical experiences of the nation's complex and turbulent history. During the Communist era, citizens were subjected to widespread state propaganda and systematic manipulation of information, eroding trust in institutions. After the collapse of communism, the shift to a market economy in 1989 brought significant challenges, including social inequalities, widespread anomie, and frustration with the inefficiency of political elites [56–58]. These historical and socio-political dynamics have left a lasting impact, fostering deep scepticism towards established institutions. This context is relevant when considering the public's perceptions of the pharmaceutical industry and healthcare. Based on the IPSOS Trustworthiness Monitor [59], only 28% of Poles trust pharmaceutical companies, compared to a global trust level of 32%. The latest IPSOS survey on trust in professions [60] reveals that this distrust extends to medical professionals, with Poland recording one of the lowest levels of trust in doctors among the 32 countries surveyed. Romania reported the lowest trust level at 40%, followed by Poland at 41% and Japan at a similar trust level. The global average was 59%.

The low levels of trust have not been improved by the restrictions on physician-industry interactions imposed by the Pharmaceutical Law [61], amended in 2007 to harmonise Polish law with Directive 2001/83/EC and Directive 2004/27/EC.

This change transformed the chaotic nature of pharmaceutical marketing into a strict law (even stricter than those required by the European Union), which, among other things, prohibits giving gifts to doctors worth more than 100 PLN (≈ 23,51 EUR). It also requires PSRs to include information consistent with the Product Information Leaflet. Only authorised individuals to issue prescriptions (e.g., physicians, some nurses) can accept drug samples after submitting a written request. The PSR must keep a record of free drugs distributed, and a doctor can receive no more than five packages of a given drug per year. According to Polish law, a pharmaceutical company may sponsor a doctor's conference fee, travel, accommodation, and meals. However, if the total value exceeds 100 PLN, they must issue a tax document for the doctor. Subsequent secondary legislation has, for example, banned PSRs from entering doctors' offices during working hours [62]. However, their enforcement is inconsistent, as evidenced in research by Makowska et al. [63].

Similarly, self-regulation by the medical profession, the Code of Medical Ethics [64,65], aims to establish ethical guidelines for the pharmaceutical industry's interactions with physicians', promoting transparency and preventing conflicts of interest. Similar initiatives exist within hospitals [66]. Additionally, the pharmaceutical industry has implemented self-regulation through the introduction of transparency codes [67,68]. Nevertheless, challenges remain in fully implementing and enforcing ethical standards in cooperation with the pharmaceutical industry across the medical community [63].

## Methods

### Study sample

A cross-sectional online survey was conducted using an internet panel. As there are around 20 million Internet users in Poland, G*Power 3.1 suggested a sample size of 968 people (alpha of 0.05, effect size of d = 0.2, allocation ratio of 0.25 for the t-test). Our actual sample was slightly higher at 1,057. The sample was selected from Adriadna Panel – one of the biggest Internet research panels in Poland, using cross-quotas, considering age, gender, and place of residence, resulting in 50 layers. Additionally, educational level and region were marginally added.

### Data collection procedure

Invitations to take part in the research were distributed and supplemented by the Ariadna panel provider to fill missing quota cells. For this reason, a conventional response rate cannot be defined here – we do not observe a denominator covering all individuals eligible to participate. Instead, we report process metrics: eligibility rate – 90.6% (1357/1497; share of eligible cases among all starts); completion rate among eligible participants (raw) – 80.5% (1093/1357; completion before cleaning among eligible cases); usable rate – 70.6% (1057/1497; completions after cleaning among all starts). Respondents received 15 points from the Ariadna panel for completing the survey; these points are accumulated across different surveys and can later be redeemed for rewards (e.g., books) The incentive was independent of response content. Data collection took place between May 16 and May 19, 2023. On average, it took around 12 minutes to complete the online survey.

### Survey instrument

The questionnaire, available through Figshare (https://doi.org/10.6084/m9.figshare.24598488), covered a range of topics related to pharmaceutical industry transparency and marketing, organised into six main sections along with 16 demographic questions. Each section included between 4 and 22 statements for respondents to evaluate. Before fielding the study in the Ariadna panel, we conducted a preliminary pilot with 12 participants. Their feedback was used to refine item wording, response options, and the questionnaire flow. In parallel, the instrument was reviewed by three experts in survey methodology, thereby supporting its content validity (adequate coverage of the construct domain) and face validity (the apparent appropriateness of the items for measuring the target construct) [69]. These steps were undertaken to ensure a sound questionnaire structure and plain, comprehensible language for respondents.

This article analysed only a few statements from sections one and three. Statements regarding trust in physicians and the pharmaceutical industry were selected from section one, each assessed on a 10-point Likert-type scale (1–10), consistent with commonly used trust measures [70,71]. From section three, we used six yes/no items that specifically examine the perception of pharmaceutical marketing presence in doctors' offices. These items were developed by the authors based on their domain expertise. To limit recall bias, we used a yes/no response format, and deliberately did not anchor items to a recent time window, as our aim was to capture whether an event had ever occurred and was memorable enough to be recalled. Accordingly, we avoided asking for frequencies or salience, which would likely have prompted reconstructive responding – something we sought to minimize. The internal consistency of this six-item measure, assessed with Cronbach's alpha, was.67, which is acceptable given the small number of items on the scale. To contextualize α, the mean inter-item correlation was.25 (range = .10–.69), which lies within the commonly recommended [72].

As the questionnaire was lengthy, we reduced respondent fatigue by using varied question formats, avoiding fully open-ended items and, where necessary, replacing them with semi-open questions. Items were grouped into logical blocks, and respondents could track their progress and proximity to completion via a progress bar.

### Ethical Approval, Informed Consent, and Reporting Guidelines

The project received approval from the Research Ethics Committee of Kozminski University no. 04/05/2023. All participants provided written informed consent. We used the STROBE cross sectional checklist when writing our report [73].

### Statistical analysis

SPSS 29 was used for statistical analysis of the data. First, the frequency of encountering different marketing activities directed at doctors was calculated. Next, the mean and standard deviation of trust in physicians and pharmaceutical companies were computed. Subsequently, t-tests were conducted to examine whether there was a difference in trust towards doctors and separately towards pharmaceutical companies among individuals who had encountered any form of marketing directed at doctors. The effect size of Cohen's d was calculated. We also calculated the number of situations each respondent encountered and examined how these situations were interconnected by creating a Pearson's bivariate correlation matrix.

As the two statements regarding meeting PSRs and waiting longer because of PSRs were correlated (close to 0.7), we decided to delete one of them. Then, we conducted linear regression analyses with the number of situations as the dependent variable. The independent variables in regression included age, gender, the presence of children in the household, the highest level of education, size of place of residence, and self-assessed health. Variables were entered into the initial model simultaneously, and their selection was guided by prior literature and the relevance to the presented topic. The model meets the basic assumptions for multiple regression [74].

### Results

The sample comprised 558 women, 497 men, and 2 individuals who identified otherwise. The mean age was 46.5 years (SD = 16), with a minimum of 18 and a maximum of 90 years. Detailed demographic characteristics of the sample are presented in Table 1.

The six forms in which respondents can encounter pharmaceutical marketing were researched. The most common method among the participants was receiving instructions from their doctors on a piece of paper with the name of the medicine (70%). As many as 44.3% noticed the logo of a pharmaceutical company in the doctor's office. More than one-third (38.1%) met a PSR in a medical facility, and 33.2% reported waiting longer due to the presence of a PSR. One quarter (25.4%) received educational materials sponsored by a pharmaceutical company from their doctor, and 23.0% received a free sample from a medical doctor.

On a 10-point Likert scale (1–10), the average trust in doctors was 6.3 (SD = 2.1), while the average trust in pharmaceutical companies was lower at 5.0 (SD = 2.3). Individuals who noticed the logo in the office, encountered a PSR in a medical

**Table 1. Distribution of key sociodemographic variables (N = 1057).**

| Feature | | % |
|---|---|---|
| Sex | Women | 52.8 |
| | Men | 47 |
| | Identified otherwise | 2 |
| Age | 18–29 | 16.8 |
| | 30–44 | 29.9 |
| | 45–59 | 27.2 |
| | 60+ | 26.1 |
| Level of education | Lower and vocational education | 20.9 |
| | Secondary education | 54.3 |
| | Bachelor's degree or higher | 24.8 |
| Place of residence | Village | 37.7 |
| | City up to 20,999 residents | 13.2 |
| | City 21,000–100,999 residents | 18.9 |
| | City 101,000–500,999 residents | 17.3 |
| | City with over 501,000 residents | 12.9 |
| Region of Poland | Kuyavian–Pomeranian | 5.3 |
| | Lublin | 5.6 |
| | Lower Silesian | 7.1 |
| | Lubusz | 2.8 |
| | Lodz | 5.6 |
| | Lesser Poland | 8.6 |
| | Masovian | 13.6 |
| | Opole | 2.9 |
| | Subcarpathian | 7.2 |
| | Podlaskie | 3.4 |
| | Pomeranian | 6.0 |
| | Silesian | 11.3 |
| | Swietokrzyskie | 3.7 |
| | Warmian–Masurian | 3.6 |
| | Greater Poland | 9.4 |
| Self-assessment of health | Very good | 7.5 |
| | Good | 38.0 |
| | Moderate | 45.2 |
| | Poor | 7.5 |
| | Very poor | 1.7 |

facility, or waited longer for an appointment due to the representative, trusted doctors statistically significantly less (see Fig 1 and Table 2 for more detail) than those who did not experience these encounters. Individuals who received a drug sample from their doctor trusted doctors statistically significantly more than those who did not receive one.

The study also found that participants who noticed a logo in the office, met a PSR in a medical facility, and experienced longer wait times for an appointment due to PSR visits had significantly lower trust in the pharmaceutical industry (Fig 2 and Table 2 have further detail) compared to those who did not have these experiences.

In addition, the number of different marketing situations each respondent encountered was calculated, including receiving instructions on branded notepads, noticing a logo in a physician's office, meeting a PSR in a physician's office,

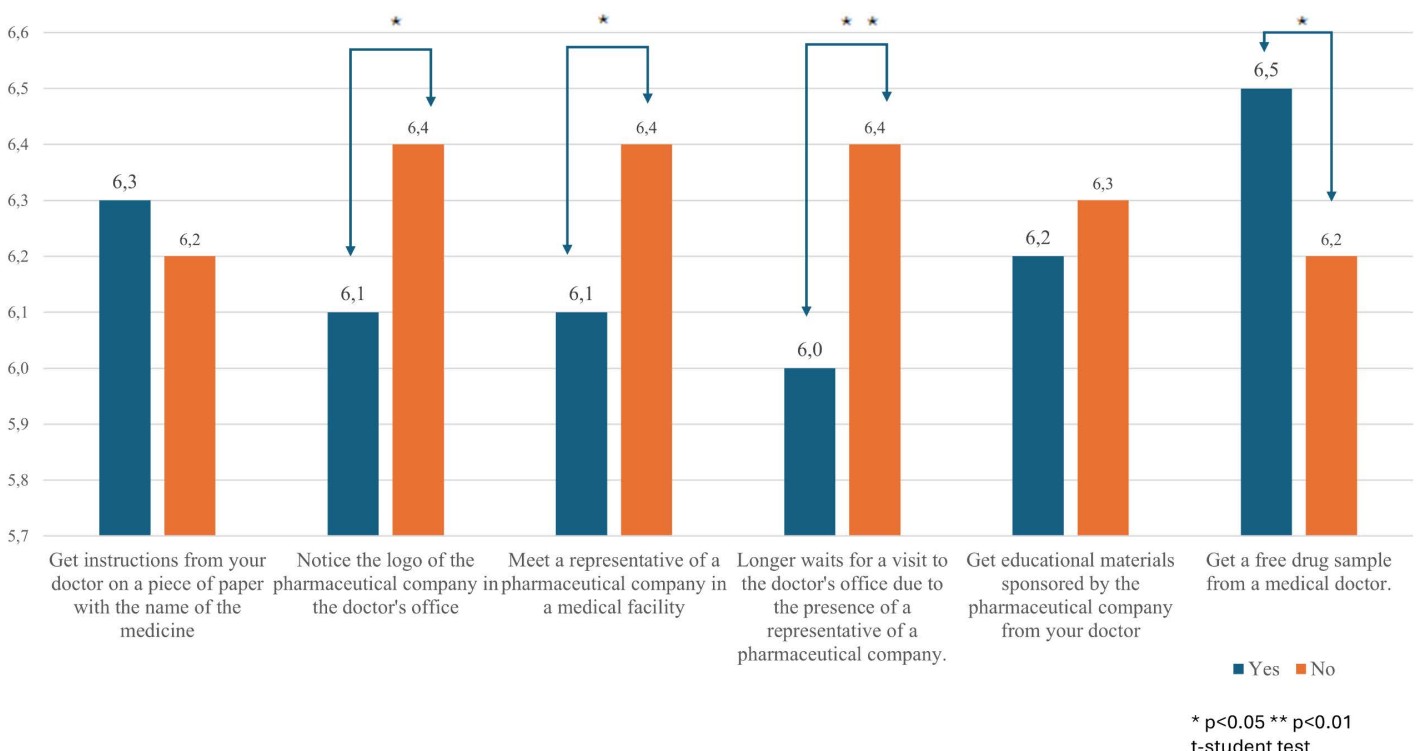

**Fig 1. Mean level of trust in physicians by self-reported exposure to pharmaceutical marketing activities.**

experiencing longer wait times due to a PSR visit, receiving educational materials, and getting a free drug sample. The interconnections between these situations were also examined. Results indicated that 16.6% of respondents had not encountered any of these situations, while an equal proportion (19.7%) had encountered one or two. Following this, 18.6% encountered three situations, 13.4% encountered four, 6.8% encountered five, and 5.5% encountered all six situations. Table 3 presents the Pearson's bivariate correlation matrix, which shows that the only strong correlation between these variables suggests that if a respondent met a PSR in a medical facility, they were also more likely to have experienced longer wait times for an appointment. Other correlations, while significant, were relatively weak.

The regression analysis indicates that the model's predictors show a weak yet statistically significant relationship with the dependent variable, which in this case is the number of experienced marketing encounters (5, as one of the correlated variables was deleted). The R-value of 0.177 indicates a modest positive correlation between the predictors and the dependent variable. The R-squared value of 0.031 suggests that only 3.1% of the variance in the dependent variable is explained by the included predictors. The Adjusted R-squared value of 0.026, which accounts for the number of predictors, indicates an even smaller proportion of explained variance, further supporting the notion that the model accounts for only a limited amount of the variability in the dependent variable.

Predictors such as being female, having at least one child in the household, and having higher-than-secondary education were associated with a slightly greater number of experienced marketing encounters. In contrast, age, size of residential area, and self-rated health were not significant predictors (see Table 4). The low explanatory power of the model suggests that these demographic variables provide only limited insight into whether patients notice physician-directed pharmaceutical marketing. This weak pattern of associations indicates that other, non-demographic determinants of such noticing should be identified in future research.

**Table 2. Relationships between self-reported exposure to pharmaceutical marketing activities and the level of trust in medical doctors and pharmaceutical companies (N = 1057).**

| Statement | | % (n) | Trust in medical doctors | | | | Trust in pharmaceutical companies | | | |
|---|---|---|---|---|---|---|---|---|---|---|
| | | | M (SD) | t (df) | p | d | M (SD) | t (df) | p | d |
| Get instructions from your doctor on a piece of paper with the name of the medicine | Yes | 70.0 (740) | 6.3 (2.1) | 1.2 (1055) | .225 | .081 | 5.0 (2.3) | −1.6 (1055) | .105 | −.108 |
| | No | 30.0 (317) | 6.2 (2.1) | | | | 5.2 (2.3) | | | |
| Notice the logo of the pharmaceutical company in the doctor's office | Yes | 44.3 (468) | 6.1 (2.1) | −2.2 (1055) | **.028** | −.136 | 4.8 (2.4) | −2.7 (966) | **.008** | −.166 |
| | No | 55.7 (589) | 6.4 (2.0) | | | | 5.2 (2.2) | | | |
| Meet a representative of a pharmaceutical company in a medical facility | Yes | 38.1 (403) | 6.1 (2.2) | −2.3 (1055) | **.019** | −.148 | 4.8 (2.4) | −3.1 (783) | **.002** | −.202 |
| | No | 61.9 (654) | 6.4 (2.0) | | | | 5.2 (2.2) | | | |
| Longer waits for a visit to the doctor's office due to the presence of a representative of a pharmaceutical company. | Yes | 33.2 (351) | 6.0 (2.2) | −2.9 (1055) | **.004** | −.188 | 4.8 (2.4) | −2.3 (1055) | **.021** | −.151 |
| | No | 66.8 (706) | 6.4 (2.0) | | | | 5.2 (2.2) | | | |
| Get educational materials sponsored by the pharmaceutical company from your doctor | Yes | 25,4 (268) | 6.2 (2.3) | −.4 (416) | .638 | −.033 | 4.8 (2.4) | −1.7 (425) | .083 | −.123 |
| | No | 74.6 (789) | 6.3 (2.0) | | | | 5.1 (2.2) | | | |
| Get a free drug sample from a medical doctor. | Yes | 23.0 (243) | 6.5 (2.2) | 2.2 (1055) | **.028** | .160 | 5.1 (2.4) | .6 (1055) | .558 | .043 |
| | No | 77.0 (814) | 6.2 (2.0) | | | | 5.0 (2.2) | | | |

## Discussion

The findings of this study highlight the significant impact of pharmaceutical marketing directed at physicians on patient trust in physicians and in the pharmaceutical industry. The results indicate that patients who encounter pharmaceutical marketing in clinical settings, such as noticing logos in doctors' offices, witnessing pharmaceutical PSRs in medical facilities, or experiencing longer wait times due to PSR visits, report significantly lower trust in both physicians and pharmaceutical companies.

Research indicates that trust is related to the appearance of the doctor [75,76], including symbols of professionalism such as a white coat [76–78]. Furthermore, the study by Kanzler and Gorsulowsky [75] highlights that the healthcare setting (private practice vs. county hospital) significantly influences patients' preferences and perceptions regarding their medical care providers. Horgan's study [79] suggests that the cleanliness and organisation of an office can also affect how its owners are perceived, which holds particular relevance in professional interactions. It is, therefore, unsurprising that the presence of a pharmaceutical company's logo in a medical office can also impact trust, as indicated by our study. What seems significant in this context is that the review by Fadlallah et al. [36] found that doctors were more accepting of educational and office-use gifts compared to personal gifts. The results of our study should encourage doctors to reflect on how any gifts from the industry can affect not only themselves but also their patients. As already noted, in Poland, there is a ban on PSRs visiting doctors during working hours [62]. Our study adds evidence to existing research [63] and suggests that this regulation is largely ineffective because 38.1% of respondents reported encountering a PSR at a medical facility, and 33.2% experienced longer wait times for their appointments because of this. Because PSRs usually avoid talking to

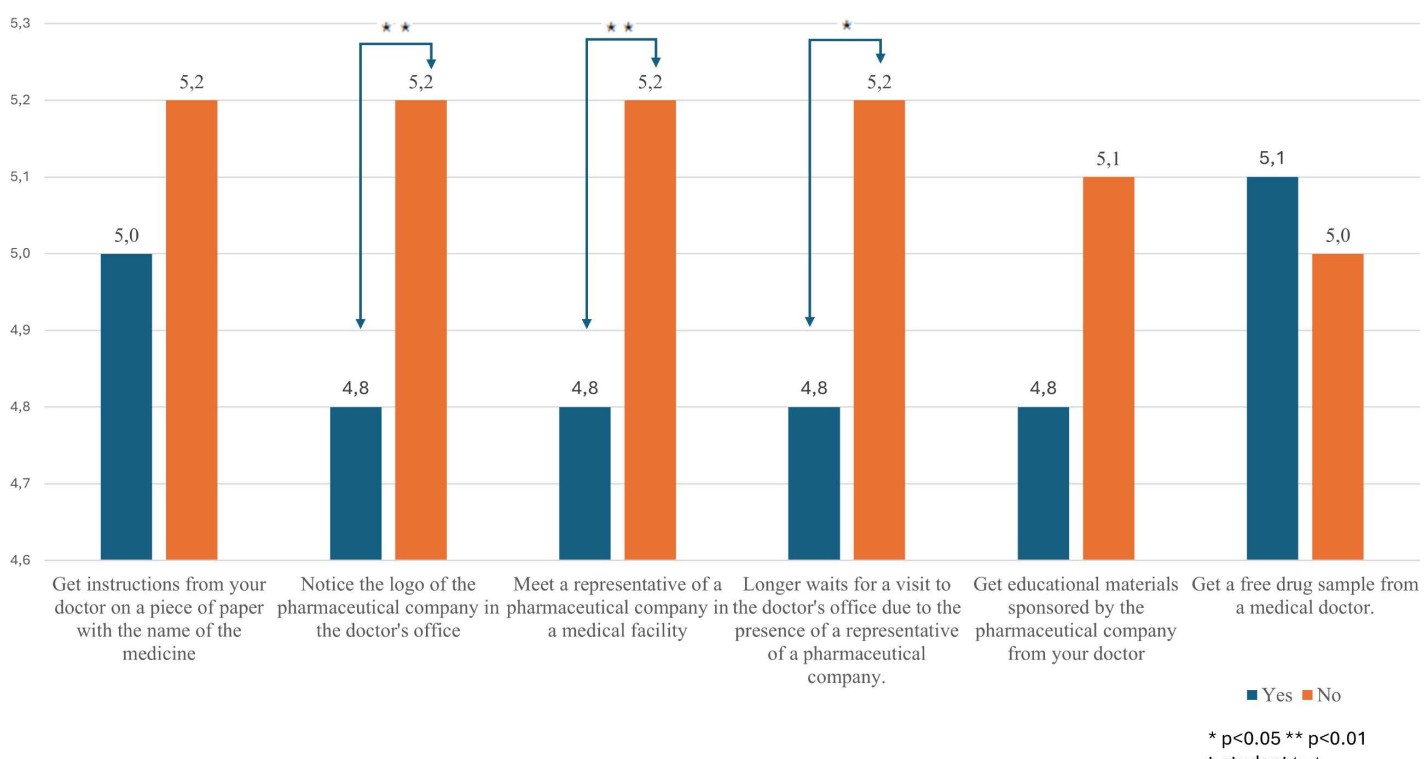

**Fig 2. Mean level of trust in the pharmaceutical industry by self-reported exposure to pharmaceutical marketing activities.**

**Table 3. Intercorrelations among self-reported encounters with pharmaceutical marketing (Pearson's r; N = 1057).**

| | Get instructions from your doctor on a piece of paper with the name of the medicine. | Notice the logo of the pharmaceutical company in the doctor's office. | Meet a representative of a pharmaceutical company in a medical facility. | Longer waits for a visit to the doctor's office due to the presence of a representative of a pharmaceutical company. | Get educational materials sponsored by the pharmaceutical company from your doctor. |
|---|---|---|---|---|---|
| Notice the logo of the pharmaceutical company in the doctor's office. | .322** | | | | |
| Meet a representative of a pharmaceutical company in a medical facility. | .225** | .300** | | | |
| Longer waits for a visit to the doctor's office due to the presence of a representative of a pharmaceutical company. | .177** | .265** | .691** | | |
| Get educational materials sponsored by the pharmaceutical company from your doctor. | .220** | .330** | 0254** | .222** | |
| Get a free drug sample from a medical doctor. | .122** | .124** | .122** | .097** | .297** |

** p < .01 (2-tailed).

**Table 4. Prediction of the number of situations encountered from socio-demographic variables (multiple regression analysis; N = 1057).**

| Independent variable | B | SE | Beta | t | p | sr* |
|---|---|---|---|---|---|---|
| Age (continious) | 0.005 | 0.003 | 0.054 | 1.639 | 0.102 | 0.050 |
| Gender (1 -female, 0-male) | 0.349 | 0.088 | 0.122 | 3.984 | **<.001** | 0.122 |
| Children in household (1-at least one, 0 – none) | 0.308 | 0.097 | 0.101 | 3.184 | **<.001** | 0.098 |
| Level of education (1 – higher than secondary 0 – secondary and lower) | 0.214 | 0.102 | 0.065 | 2.091 | **0.037** | 0.064 |
| Size of residential area (1 – city over 501K 0- village and city up to 500K) | 0.063 | 0.131 | 0.015 | 0.485 | 0.628 | 0.015 |
| Self-assessment of health status (1 – very good, good; 0 – moderate, poor, very poor) | 0.136 | 0.091 | 0.047 | 1.495 | 0.135 | 0.046 |
| Constant | 1.371 | 0.159 | | 8.638 | **<.001** | |

*Semi-partial correlation (sr) is a statistical measure used in multiple regression analyses to assess the distinct impact of each predictor variable on the dependent variable, accounting for the effects of other predictors.

physicians in hallways or with doors open – practices that are considered illegal – they may still be present in ways that are difficult to detect. However, attentive patients can pick up indications of PSR presence: 1) a representative 'cutting the line' while explaining they are 'just for a moment'; 2) entering the doctor's office together with a patient – 'just to show that I am here' – and then being explicitly invited in from the corridor by the physician; 3) walking from one consulting room to another, sometimes using corridors normally reserved for staff; and 4) wearing distinctive business attire or carrying promotional materials [54,80,81]. Such violations of the law by doctors do not contribute to building patients' trust in them. However, we do not know whether this trust diminishes because patients are aware that doctors are breaking the law by accepting visits from PSRs or because merely seeing the representative raises suspicions about the doctor's impartiality. This issue certainly requires further research.

Interestingly, receiving free drug samples from physicians was the only marketing activity associated with increased trust in physicians. This suggests that patients might perceive free samples as a direct benefit to themselves rather than as evidence of physician-industry entanglement. In this regard, rather than interpreting this as respondents accepting free samples, it would be important to consider to what extent patients contextualise the significance of such interactions. Research clearly shows that free drug samples are a powerful tool for pharmaceutical marketing [82–84]. They work similarly to other gifts by triggering the rule of reciprocity mechanism [11], causing doctors to feel compelled (often unconsciously) to 'repay' the company that provided the gift, especially when the gifts align with the doctors' own purposes [85]. Drug samples are also significant for drug company marketing for another important reason – patients receiving these samples may start long–time treatment with more expensive medications, even when cheaper alternatives are available [86]. In Poland, there are also restrictions on PSRs providing drug samples to doctors. Samples can only be given upon written request, and a maximum of five of the smallest packages can be received per year [61]. However, previous studies have found that these regulations are not always followed [80].

The presented results are likely to reflect Polish people's low level of trust in medical professionals and the pharmaceutical industry. The low baseline trust in physicians and pharmaceutical companies reflects broader societal attitudes towards authority and institutions, rooted in historical and socio-political context [56], which must also be recognised by doctors who wish to treat their patients effectively.

Our regression model linking demographic factors to the detection of pharmaceutical marketing exhibited low explanatory power, and the observed associations had small effect sizes. This suggests that socio-demographic characteristics alone have limited predictive value in this dataset and specification. More promising predictors may include: attitudes toward the industry, health literacy, health anxiety, locus of control, exposure to information sources, and the patient–physician relationship, which were not included here. These considerations point to important directions for future research.

Although previous studies have documented associations between socio-demographic factors and selective attention [39,46–51], these findings may not directly translate to the detection of pharmaceutical marketing during medical visits.

The latter is a more complex, context-dependent behaviour, shaped not only by basic attentional mechanisms but also by patients' immediate focus on their health and the doctor's advice during the consultation.

## Policy and practice implications and recommendations

This erosion of trust may have far-reaching implications for public health. Patients may become reluctant to take prescribed medications, refuse recommended or mandatory vaccinations, and disregard medical advice [87–89]. This distrust can lead to the proliferation of conspiracy theories about healthcare and increased reliance on unproven alternative medicine [90], ultimately resulting in worse health outcomes and undermining patient safety. Previous research aligns with these findings, showing that trust in the healthcare system is fragile and can easily be undermined by perceptions of commercial influence [22,37]. However, the observed contextual factors highlight the need for country-specific interventions. Therefore, policy approaches developed in Western European or US contexts may not yield the same outcomes in Poland without accounting for cultural nuances and historical distrust in institutions. Similar patterns have been observed in other fields throughout Poland's economic transition to capitalism and liberal democracy, including international aid [91].

Several practical recommendations emerge from this study to address the challenges of physician-pharmaceutical company interactions. First, healthcare providers should implement comprehensive training programs and continuing medical education courses to raise physicians' awareness of the potential consequences of pharmaceutical collaborations, particularly focusing on recognising, resisting and challenging potentially manipulative marketing techniques [92]. Second, there should be increased emphasis on promoting and enforcing transparency legislation regarding physician-pharmaceutical company interactions. Third, existing Polish regulations prohibiting meetings between PSRs and physicians during working hours should be more strictly enforced. Finally, efforts should be made to minimise the presence of pharmaceutical promotional materials in doctors' offices, although this will be challenging to implement in resource-constrained settings, particularly in underfunded public clinics. These recommendations aim to maintain professional independence while acknowledging the practical constraints faced by healthcare institutions.

## Limitations

Despite using cross-quotas based on age, gender, and place of residence to select the sample, the study relies on an internet panel, which may lead to sampling bias. Panel participants may have specific characteristics that motivate them to complete surveys in exchange for rewards. This self-selection into the panel, together with heterogeneous response propensities among panellists, can introduce systematic biases. Not all individuals in Poland have internet access [93], and certain groups – such as those from lower socio-economic backgrounds, older individuals, or rural residents – may be underrepresented in online surveys. Taken together, these factors could limit the ability to generalise the findings to the broader population. Furthermore, since the study focuses on the experiences of participants from Poland, the results may not apply to other countries with different cultural, social, or healthcare contexts. Therefore, the findings should be considered within the study's specific context. Finally, because we asked about ever-observed events (rather than a specific recent visit), some respondents may have failed to recall their experiences; in such cases, 'no' responses would underestimate prevalence and bias associations toward zero. In addition, dichotomous items do not capture frequency or salience (e.g., a prominent logo or a persistent conversation with a PSR with the door open), further reducing information.

## Future research

This study illuminates important aspects of pharmaceutical marketing and its impact on trust in healthcare systems, but several critical areas warrant further investigation. Future research should examine the long-term and diverse consequences of pharmaceutical marketing encounters on patient health outcomes and treatment adherence, utilising longitudinal studies to track these relationships over time.

Methodologically, future studies could anchor questions to a bounded recall window and replace binary indicators of marketing visibility with simple frequency or salience scales. Alternatively, experimental designs could be used to determine whether marketing cues were in fact noticed.

Subsequent analyses should also incorporate richer psychological and relational constructs (e.g., attitudes toward the industry, health literacy, health anxiety, locus of control, exposure to information sources, and the patient–physician relationship). By contrast, conventional sociodemographic variables, which explained little in our data, appear to offer limited explanatory value on their own.

Additionally, researchers should investigate perceptions of pharmaceutical marketing directed at physicians, incorporating qualitative research methods such as in-depth interviews and focus groups to provide a richer contextual understanding of patient experiences and perspectives. Finally, empirical studies are needed to evaluate the effectiveness of specific regulatory interventions in enhancing patient trust, particularly focusing on measurable outcomes and implementation challenges in various healthcare settings. Such research would provide valuable insights for developing evidence-based strategies to optimise the balance between pharmaceutical industry engagement and public trust in healthcare systems.

## Conclusion

This study reveals important insights into the relationship between doctor-targeted pharmaceutical marketing and patient trust in healthcare providers. Our findings demonstrate that while overall trust in physicians remains moderate, trust in pharmaceutical companies is notably lower. Pharmaceutical marketing elements observed by patients – especially those that affect access to the doctor or prolong waiting times – undermine trust. By contrast, free drug samples may be perceived as a direct patient benefit, partly offsetting negative associations with marketing. Healthcare providers and pharmaceutical companies must recognise that their marketing relationships have tangible effects on patient trust and carefully consider the implications of their collaborations for maintaining public confidence in the healthcare system. In particular, adherence to Polish law regulating pharmaceutical sales representatives' access to physicians, as well as the removal of visible marketing elements from patient-facing areas, is advisable.

## Author contributions

**Conceptualization:** Marta Makowska.

**Formal analysis:** Marta Makowska.

**Funding acquisition:** Marta Makowska.

**Methodology:** Marta Makowska.

**Supervision:** Marta Makowska.

**Writing – original draft:** Marta Makowska, Akihiko Ozaki, Piotr Ozieranski.

**Writing – review & editing:** Marta Makowska, Akihiko Ozaki, Piotr Ozieranski.

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
