## [Decision Letter · Decision Letter 0]

29 Oct 2025

Dear Dr. Ozieranski,

It is not clear how long passed between the time of the respondents attending a physician office visit and the time they completed the survey.  This brings up the concern of recall bias by respondents.  Please address how this concern was minimized.It is not clear to me that a patient in a physician office would be able to recognize a pharmaceutical sales representative.  How can we be sure that the responses accurately reflect what respondents witnessed?Please also identify currency in the equivalent in Euros or Dollars.  Most readers will not be familiar with the value of Polish currency.In the methods, it is stated, "Respondents were rewarded by points from the panel....."  It is not clear what this means. What were these points?  What does it mean to be rewarded with points?The survey instrument appears to be quite long.  How was survey fatigue by respondents addressed?Why use a 10 point Likert scale?  Four, 5 or 7, points would be more common.

Please submit your revised manuscript by Dec 13 2025 11:59PM.  If you will need more time than this to complete your revisions, please reply to this message or contact the journal office at plosone@plos.org . . A rebuttal letter that responds to each point raised by the academic editor and reviewer(s). You should upload this letter as a separate file labeled 'Response to Reviewers'.A marked-up copy of your manuscript that highlights changes made to the original version. You should upload this as a separate file labeled 'Revised Manuscript with Track Changes'.An unmarked version of your revised paper without tracked changes. You should upload this as a separate file labeled 'Manuscript'.

We look forward to receiving your revised manuscript.

Kind regards,

John Rovers, PharmD, MIPH

Academic Editor

PLOS ONE

Journal Requirements:

[Funding was received by Dr Makowska from the Kozminski University, Warsaw Poland].

3. Thank you for stating the following in your manuscript:

[This research received funding in the form of ‘small grant’ from Kozmininski University]

[Funding was received by Dr Makowska from the Kozminski University, Warsaw Poland]

4. Thank you for providing the following Competing Interests Statement:

[MM nothing to declare; AO has received the following: Consulting fees from: Beckton, Dickinson and Company, MNES Inc. Payment or honoraria for lectures, presentations, speakers bureaus, manuscript writing, or educational events from:

Pfizer, Daiichi Sankyo, Taiho Pharmaceutical Co. Ltd., Kyowa Kirin. PO’s former PhD student was supported by a grant from Sigma Pharmaceuticals, a UK pharmacy wholesaler and distributor (not a pharmaceutical company). The PhD work funded by Sigma Pharmaceuticals is unrelated to the subject of this paper.].

We note that one or more of the authors is affiliated with the funding organization, indicating the funder may have had some role in the design, data collection, analysis or preparation of your manuscript for publication; in other words, the funder played an indirect role through the participation of the co-authors.

If the funding organization did not play a role in the study design, data collection and analysis, decision to publish, or preparation of the manuscript and only provided financial support in the form of authors' salaries and/or research materials, please review your statements relating to the author contributions, and ensure you have specifically and accurately indicated the role(s) that these authors had in your study in the Author Contributions section of the online submission form. Please make any necessary amendments directly within this section of the online submission form.  Please also update your Funding Statement to include the following statement: “The funder provided support in the form of salaries for authors [insert relevant initials], but did not have any additional role in the study design, data collection and analysis, decision to publish, or preparation of the manuscript. The specific roles of these authors are articulated in the ‘author contributions’ section.”

If the funding organization did have an additional role, please state and explain that role within your Funding Statement.

Please also provide an updated Competing Interests Statement declaring this commercial affiliation along with any other relevant declarations relating to employment, consultancy, patents, products in development, or marketed products, etc.

Reviewers' comments:

Reviewer's Responses to Questions

1. Is the manuscript technically sound, and do the data support the conclusions?

Reviewer #1: Partly

Reviewer #2: Partly

2. Has the statistical analysis been performed appropriately and rigorously?

Reviewer #1: Yes

Reviewer #2: Yes

3. Have the authors made all data underlying the findings in their manuscript fully available?

Reviewer #1: Yes

Reviewer #2: Yes

4. Is the manuscript presented in an intelligible fashion and written in standard English?

Reviewer #1: Yes

Reviewer #2: Yes

Reviewer #1: By cleverly shifting the focus from the physician to the patient and placing the research in the unique Polish cultural-legal context, the authors have made an original scientific contribution. However, the significance of their findings is limited by a methodological flaw: Internet sampling. The limitations section of the paper should therefore discuss this issue and warn about generalizability.

In the abstract findings, use statistical data such as regression coefficients or p-values.

The introduction of the article presents a one-sided and negative picture of the pharmaceutical industry and describes physician-industry interactions merely as risky marketing behaviors, which indicates the bias of the authors and reduces the scientific credibility of the work. A paragraph should be added to the introduction that also points out the positive and necessary roles of these interactions (such as education, exchange of scientific information, and support of research) to make the article look balanced and professional. Please consider this article in the introduction also:

“Interaction between physicians and the pharmaceutical industry: A scoping review for developing a policy brief”

https://www.frontiersin.org/journals/public-health/articles/10.3389/fpubh.2022.1072708/full

The article states that trust in doctors in Poland is low (in global comparison), but at the same time they are the second most trusted group in the country. This apparent contradiction may confuse the reader.

I recommend dividing the methodology into separate sections such as design and sampling, data collection tools and processes, and data analysis.

More details about the questionnaire are needed and should be included. For example, how the questions were designed and references cited.

Lack of validation of the questionnaire: The authors do not indicate whether the questionnaire (especially the six items on marketing perception) was validated before use. Was a pilot study conducted to ensure the clarity and understandability of the questions? Was Cronbach’s alpha used to measure the internal reliability of the items? This is an important gap in the methodological report.

Nature of the questions: The questions on marketing observation were asked as “yes/no”. This method is simple but misses the point. For example, the patient cannot say “how many times” they saw a representative or how “salient” the observation was. This may reduce the power of the analysis.

There is need to detail about sampling procedures: How were people notified to participate in the study? Response rate? And other details

The titles of the graphs and tables are not descriptive.

Limitations of the predictive model: The weakness of the regression model is an important lesson: simply knowing a patient’s age and gender does not tell us how much they will pay attention to marketing. This suggests that deeper psychological and situational variables need to be addressed to fully understand this phenomenon.

The absence of a demographic table makes it very difficult for the reader to assess the external validity and generalizability of the results and reduces the scientific strength of the article. There needs to be such a table with brief descriptions at the beginning of the findings section.

An ideal conclusion should avoid statistical details and focus on conceptual and strategic messages.

Reviewer #2: thank you for this interesting work. The information on the impact of each factor on causing it to be noticed is interesting, and they effects on how the patients view each type of promotional action is well done.

i would suggest limiting your discussion on the low r2 of the regression and admit it wasn't a great model, instead of the in-depth analysis.

Does your background theory suggest any other possible model for which you have measures? It the dependent variable really the one that you want to know about? Think about another regression equation or minimize the discussion of that measure.

**Do you want your identity to be public for this peer review?** For information about this choice, including consent withdrawal, please see our For information about this choice, including consent withdrawal, please see our Privacy Policy .

Reviewer #1: No

Reviewer #2: No

---

## [Author Response · Author response to Decision Letter 1]

17 Nov 2025

Thank you for overseeing the review of our manuscript, ‘Pharmaceutical Marketing and Patient Trust: How Do Doctor-Targeted Campaigns Affect Patients?’ We are grateful for your additional supportive comments and the reviewers’ thoughtful reports. We have addressed every point of your review below.

Remark#1

It is not clear how long passed between the time of the respondents attending a physician office visit and the time they completed the survey. This brings up the concern of recall bias by respondents. Please address how this concern was minimized.

Thank you for drawing our attention to the issue of recall bias. Indeed, it had not been fully discussed in the limitations section – we have now addressed this gap. To minimize the impact of memory lapses, we did not ask respondents to reconstruct details of their most recent doctor’s visit; instead, we asked whether they had ever experienced specific events. Thus, we relied on recognition of an event rather than precise episodic memory of a particular visit; consequently, the time elapsed between the visit and completing the survey was not analytically crucial in this context. Specifically: 1) we used brief, behaviorally specific, and neutral wording (e.g., “Have you ever: b) noticed the logo of a pharmaceutical company in the doctor’s office?”), 2) we employed mutually exclusive response categories – yes/no, and 3) we avoided requiring counts of events or their dating. Despite these steps, complete elimination of recall bias is not possible; however, if it occurred, it was most likely non-differential and therefore would tend to attenuate rather than inflate the observed associations. We have added this information to the Methods, Limitations and Future Research sections. The fragments are as follows:

Methods:

‘From section three, we used six yes/no items that specifically examine the perception of pharmaceutical marketing presence in doctors’ offices. These items were developed by the authors based on their domain expertise. To limit recall bias, we used a yes/no response format, and deliberately did not anchor items to a recent time window, as our aim was to capture whether an event had ever occurred and was memorable enough to be recalled. Accordingly, we avoided asking for frequencies or salience, which would likely have prompted reconstructive responding – something we sought to minimize. The internal consistency of this six-item measure, assessed with Cronbach’s alpha, was .67, which is acceptable given the small number of items on the scale. To contextualize α, the mean inter-item correlation was .25 (range = .10–.69), which lies within the commonly recommended [72].’

Limitations:

‘Finally, because we asked about ever-observed events (rather than a specific recent visit), some respondents may have failed to recall their experiences; in such cases, ‘no’ responses would underestimate prevalence and bias associations toward zero. In addition, dichotomous items do not capture frequency or salience (e.g., a prominent logo or a persistent conversation with an open door), further reducing information.’

Future Research:

‘Methodologically, future studies could anchor questions to a bounded recall window and replace binary indicators of marketing visibility with simple frequency or salience scales. Alternatively, experimental designs could be used to determine whether marketing cues were in fact noticed.’

Remark#2

It is not clear to me that a patient in a physician office would be able to recognize a pharmaceutical sales representative. How can we be sure that the responses accurately reflect what respondents witnessed?

Thank you for this question. Previous research conducted by Dr. Makowska (2010) targeting pharmaceutical sales representatives (PSRs) documented the strategies they use to gain access to physicians’ offices in Poland. Many of these practices are visible to patients and can sometimes provoke negative reactions – from being called out to occasional insults. In the Polish system, patients are usually scheduled for specific appointment times, so classic queues form less often; PSRs are unscheduled, do not wait in line, and instead try to bypass it. Although some PSRs consciously try to minimize their visibility and ‘blend in’ with patients to avoid tensions in the waiting room, this is not always the case; at times they approach physicians in the corridor or hold conversations with the door ajar. Not all patients will notice representatives, and not all representatives are easy to spot. Nevertheless, some patients are able to pick up on these signals, and this is precisely what we set out to examine. A paragraph was added to the Discussion:

‘Because PSRs usually avoid talking to physicians in hallways or with doors open – practices that are considered illegal – they may still be present in ways that are difficult to detect. However, attentive patients can pick up indications of PSR presence: 1) a representative ‘cutting the line’ while explaining they are ‘just for a moment’; 2) entering the doctor’s office together with a patient – ‘just to show that I am here’ – and then being explicitly invited in from the corridor by the physician; 3) walking from one consulting room to another, sometimes using corridors normally reserved for staff; and 4) wearing distinctive business attire or carrying promotional materials [54,80,81]’

Remark#3

Please also identify currency in the equivalent in Euros or Dollars. Most readers will not be familiar with the value of Polish currency.

In the manuscript, all amounts were converted to euros. Moreover, we updated these figures when new data from the Central Statistical Office became available a few months ago. The text now reads as follows:

‘According to the Central Statistical Office [55], 2024 marked a record year for prescription drug sales in Poland, both in volume and value. The number of prescriptions issued rose by 3.7% compared to the previous year. Furthermore, the total value of prescription drugs purchased reached 29.6 billion PLN (≈ 6,96bn EUR; exchange rate from 3 November 2025: 1 EUR = 4.25 PLN) reflecting a year-on-year growth of 3.5 billion PLN (≈ 0,82bn EUR).’

‘This change transformed the chaotic nature of pharmaceutical marketing into a strict law (even stricter than those required by the European Union), which, among other things, prohibits giving gifts to doctors worth more than 100 PLN (≈ 23,51 EUR).’

Remark#4

In the methods, it is stated, "Respondents were rewarded by points from the panel....." It is not clear what this means. What were these points? What does it mean to be rewarded with points?

In Poland, online research panels maintain a database of panelists with diverse socio-demographic characteristics, which enables the construction of appropriate samples for studies. Participants do not receive direct monetary compensation; instead, points are awarded for completing questionnaires, which can be exchanged for rewards offered by the panel (e.g., books, lego blocks). The value of the incentive depends on survey completion and the expected time commitment, not on the content of the responses. A sentences was added to the manuscript:

‘Respondents received 15 points from the Ariadna panel for completing the survey; these points are accumulated across different surveys and can later be redeemed for rewards (e.g., books). The incentive was independent of response content.’

Remark#5

The survey instrument appears to be quite long. How was survey fatigue by respondents addressed?

We have addressed Your comment by adding the following sentence to the manuscript:

‘As the questionnaire was lengthy, we reduced respondent fatigue by using varied question formats, avoiding fully open-ended items and, where necessary, replacing them with semi-open questions. Items were grouped into logical blocks, and respondents could track their progress and proximity to completion via a progress bar.’

Remark#6

Why use a 10 point Likert scale? Four, 5 or 7, points would be more common.

Trust was measured using a 10-point Likert-type scale (1–10). This choice is consistent with commonly used trust measures (Belgian Health Care Knowledge Centre, 2024; European Foundation for the Improvement of Living and Working Conditions, 2025). OECD guidelines on measuring trust also use a 0–10 response format (OECD, 2017, p. 17). We adopted the 1–10 format to ensure consistency with the response scales used for other items in the questionnaire. Prior work also shows that 10-point scales are preferred by respondents and facilitate finer discrimination (Preston & Colman, 2000). It is now indicated in the manuscript that we used a 1–10 scale, not 0–10. The text is now as follows:

‘Statements regarding trust in physicians and the pharmaceutical industry were selected from section one, each assessed on a 10-point Likert-type scale (1–10), consistent with commonly used trust measures [70,71].’

We hope that these changes are acceptable. If you have any further comments we would be happy to address them.

Sincerely,

Authors

References:

Belgian Health Care Knowledge Centre. (2024, February 1). Performance of the Belgian health system: Report 2024. https://kce.fgov.be/en/performance-of-the-belgian-health-system-report-2024

European Foundation for the Improvement of Living and Working Conditions. (2025). Trust in institutions, EU, 2020–2024 (scale 1–10). Eurofound. https://www.eurofound.europa.eu/en/surveys-and-data/data-catalogue/trust-institutions-eu-2020-2024-scale-1-10

OECD. (2017). OECD Guidelines on Measuring Trust. OECD Publishing. https://doi.org/10.1787/9789264278219-en

Preston, C. C., & Colman, A. M. (2000). Optimal number of response categories in rating scales: Reliability, validity, discriminating power, and respondent preferences. Acta Psychologica, 104(1), 1–15. https://doi.org/10.1016/s0001-6918(99)00050-5

Dear Reviewer 1,

Thank you for offering a very detailed and generous review of our article ‘‘Pharmaceutical Marketing and Patient Trust: How Do Doctor-Targeted Campaigns Affect Patients?’. We have sought to address your points as comprehensively as possible. Our responses to your specific comments are provided below.

Remark #1

By cleverly shifting the focus from the physician to the patient and placing the research in the unique Polish cultural-legal context, the authors have made an original scientific contribution. However, the significance of their findings is limited by a methodological flaw: Internet sampling. The limitations section of the paper should therefore discuss this issue and warn about generalizability.

Thank you for this comment. We elaborated on this point in the limitations section as follows:

‘Despite using cross-quotas based on age, gender, and place of residence to select the sample, the study relies on an internet panel, which may lead to sampling bias. Panel participants may have specific characteristics that motivate them to complete surveys in exchange for rewards. This self-selection into the panel, together with heterogeneous response propensities among panellists, can introduce systematic biases. Not all individuals in Poland have internet access [93], and certain groups – such as those from lower socio-economic backgrounds, older individuals, or rural residents – may be underrepresented in online surveys. Taken together, these factors could limit the ability to generalise the findings to the broader population.’

Remark #2

In the abstract findings, use statistical data such as regression coefficients or p-values.

The abstract has been revised in accordance with the suggestions. The Results section now reads as follows:

‘Results: The average trust in physicians among Poles, on a 10-point scale, was 6.3 (SD = 2.1), while the average trust in pharmaceutical companies was lower, at 5.0 (SD = 2.3). The results indicate that 83.4% of respondents noticed signs of pharmaceutical marketing directed at doctors, with 5.5% experiencing all six types of marketing practices addressed in the study. Seeing a company logo in the doctor’s office, encountering a pharmaceutical sales representative (PSR), and experiencing PSR-related longer waits were each associated with lower trust in physicians (t = −2.2, −2.3, −2.9; p = .028, .019, .004; d = −.136, −.148, −.188 , respectively) and in pharmaceutical companies (t = −2.7, −3.1, −2.3; p = .008, .002, .021; d = −.166, −.202, −.151 , respectively). Receiving a free drug sample was linked to slightly higher trust in physicians (t = 2.2, p = .028, d = .16) and showed no effect on trust in companies (p = .558). Most pairwise correlations among patient-encounter pharmaceutical marketing situations were weak, even when they reached statistical significance; the only strong association was between encountering a PSR in a medical facility and reporting PSR-related longer wait times (r = .69, p < .001).’

Remark #3

The introduction of the article presents a one-sided and negative picture of the pharmaceutical industry and describes physician-industry interactions merely as risky marketing behaviors, which indicates the bias of the authors and reduces the scientific credibility of the work. A paragraph should be added to the introduction that also points out the positive and necessary roles of these interactions (such as education, exchange of scientific information, and support of research) to make the article look balanced and professional. Please consider this article in the introduction also: “Interaction between physicians and the pharmaceutical industry: A scoping review for developing a policy brief”

In line with the reviewer’s suggestion, we added a paragraph outlining the beneficial aspects of pharmaceutical marketing. We also appreciate the recommended article, which has been incorporated into our argumentation. The added paragraph reads as follows:

‘Yet, it is worth noting that the marketing activities of the pharmaceutical industry also contain elements that may be beneficial from the perspective of the health system and patients [28,29]. Given the relative dearth of alternative funding sources, the pharmaceutical industry offers professional development opportunities, enabling physicians to regularly update their knowledge and participate in conferences and training [30–32] and, with appropriate safeguards, this education need not be biased [31]. PSRs frequently provide drug samples that can be given to patients to try a medication at no cost [30,33]. Physicians may perform various consultancies and research for industry, which constitute an important source of additional income for some doctors or contribute to the development and testing of new medicines [32,34].’

Remark #4

The article states that trust in doctors in Poland is low (in global comparison), but at the same time they are the second most trusted group in the country. This apparent contradiction may confuse the reader.

We are grateful for this observation; following the reviewer’s comment, we deleted the confusing information.

Remark#5

I recommend dividing the methodology into separate sections such as design and sampling, data collection tools and processes, and data analysis.

We have reorganized the Methods section in line with the reviewer’s suggestions; please see the revised Methods section in manuscript.

Remark#6

More details about the questionnaire are needed and should be included. For example, how the questions were designed and references cited. Lack of validation of the questionnaire: The authors do not indicate whether the questionnaire (especially the six items on marketing perception) was validated before use. Was a pilot study conducted to ensure the clarity and understandability of the questions? Was Cronbach’s alpha used to measure the internal reliability of the items? This is an important gap in the methodological report.

Additional information about the questionnaire has been included. Where the question format was inspired by the literature, this has been indicated. Information on the instrument’s validation has also been added. The text now reads as follows:

‘Before fielding the study in the Ariadna panel, we conducted a preliminary pilot with 12 particip

---

## [Decision Letter · Decision Letter 1]

22 Feb 2026

Pharmaceutical Marketing and Patient Trust: How Do Doctor-Targeted Campaigns Affect Patients

PONE-D-25-23673R1

Dear Dr. Ozieranski

We’re pleased to inform you that your manuscript has been judged scientifically suitable for publication and will be formally accepted for publication once it meets all outstanding technical requirements.

Kind regards,

John Rovers, PharmD, MIPH

Academic Editor

PLOS One

Additional Editor Comments (optional):

Reviewers' comments:

Reviewer's Responses to Questions

**Comments to the Author**

Reviewer #1: All comments have been addressed

Reviewer #2: All comments have been addressed

2. Is the manuscript technically sound, and do the data support the conclusions?

Reviewer #1: Yes

Reviewer #2: Yes

3. Has the statistical analysis been performed appropriately and rigorously?

Reviewer #1: Yes

Reviewer #2: Yes

4. Have the authors made all data underlying the findings in their manuscript fully available?

Reviewer #1: Yes

Reviewer #2: Yes

5. Is the manuscript presented in an intelligible fashion and written in standard English?

Reviewer #1: Yes

Reviewer #2: Yes

Reviewer #1: no comments no comments no comments no comments no comments no comments no comments no comments no comments no comments

Why are you forcing a comment? Why does the reviwer need to write 100 words when he doesn't have a comment?

Reviewer #2: thank you for putting so much effort into these revisions. They address all of the significant issues identified by all reviewers in a very professional manner.

**Do you want your identity to be public for this peer review?** For information about this choice, including consent withdrawal, please see our For information about this choice, including consent withdrawal, please see our Privacy Policy .

Reviewer #1: No

Reviewer #2: No

---

## [Editor Report · Acceptance letter]

PONE-D-25-23673R1

PLOS One

Dear Dr. Ozieranski,

I'm pleased to inform you that your manuscript has been deemed suitable for publication in PLOS One. Congratulations! Your manuscript is now being handed over to our production team.

Kind regards,

on behalf of

Dr. John Rovers

Academic Editor

PLOS One